# Design and Experiment of a Visual Detection System for Zanthoxylum-Harvesting Robot Based on Improved YOLOv5 Model

**Jinkai Guo [1,2], Xiao Xiao [1,2], Jianchi Miao [1,2], Bingquan Tian [1,2], Jing Zhao [1,2,*] and Yubin Lan [3,*]**

[1] School of Agricultural Engineering and Food Science, Shandong University of Technology, Zibo 255000, China
[2] National Sub-Center for International Collaboration Research on Precision Agricultural Aviation Pesticide Spraying Technology, Shandong University of Technology, Zibo 255000, China
[3] College of Electronic Engineering, South China Agricultural University, Guangzhou 510642, China
* Correspondence: zhaojing@sdut.edu.cn (J.Z.); ylan@scau.edu.cn (Y.L.)

**Abstract:** In order to achieve accurate detection of mature Zanthoxylum in their natural environment, a Zanthoxylum detection network based on the YOLOv5 object detection model was proposed. It addresses the issues of irregular shape and occlusion caused by the growth of Zanthoxylum on trees and the overlapping of Zanthoxylum branches and leaves with the fruits, which affect the accuracy of Zanthoxylum detection. To improve the model's generalization ability, data augmentation was performed using different methods. To enhance the directionality of feature extraction and enable the convolution kernel to be adjusted according to the actual shape of each Zanthoxylum cluster, the coordinate attention module and the deformable convolution module were integrated into the YOLOv5 network. Through ablation experiments, the impacts of the attention mechanism and deformable convolution on the performance of YOLOv5 were compared. Comparisons were made using the Faster R-CNN, SSD, and CenterNet algorithms. A Zanthoxylum harvesting robot vision detection platform was built, and the visual detection system was tested. The experimental results showed that using the improved YOLOv5 model, as compared to the original YOLOv5 network, the average detection accuracy for Zanthoxylum in its natural environment was increased by 4.6% and 6.9% in terms of mAP@0.5 and mAP@0.5:0.95, respectively, showing a significant advantage over other network models. At the same time, on the test set of Zanthoxylum with occlusions, the improved model showed increased mAP@0.5 and mAP@0.5:0.95 by 5.4% and 4.7%, respectively, compared to the original model. The improved model was tested on a mobile picking platform, and the results showed that the model was able to accurately identify mature Zanthoxylum in its natural environment at a detection speed of about 89.3 frames per second. This research provides technical support for the visual detection system of intelligent Zanthoxylum-harvesting robots.

**Keywords:** YOLOv5; deformable convolution; attention mechanism; visual detection system; Zanthoxylum-harvesting robot

## 1. Introduction

Zanthoxylum is widely cultivated in various parts of China, with a cultivated area of about 17.284 million mu and an annual output of over 500,000 tons. It is an important medicinal material and food ingredient. However, manual harvesting of Zanthoxylum is faced with the problems of low efficiency, high temperature and humidity in the work environment, severe mosquito bites, and injuries to workers. Developing a Zanthoxylum-harvesting robot can reduce labor intensity and improve the efficiency of harvesting [1–3].

To enhance the efficiency and quality of Zanthoxylum harvesting, researchers have studied the mechanical harvesting of Zanthoxylum. Wan Fangxin and others designed a comb-type Zanthoxylum harvester using the principles of brushing and air suction [4].

Zheng Tianyun designed an electromagnetic Zanthoxylum picker [5]. During the development of the Zanthoxylum harvester, it was intended to be lightweight and easy to operate, with semi-automated portability [6]. Although these studies have, to some extent, improved the efficiency of Zanthoxylum harvesting and reduced the labor intensity, the lack of an automatic recognition and positioning system for Zanthoxylum makes it difficult to achieve fully automatic harvesting [7,8]. Many scholars have proposed target detection and localization algorithms for Zanthoxylum based on computer vision. Zhang Yongmei and others used a combination of color analysis and image fusion algorithms to identify Zanthoxylum [9]. Yang Ping et al. used the K-means clustering algorithm for target extraction of Zanthoxylum [10]. As deep learning technology advances, the application of object detection algorithms in agriculture is becoming increasingly widespread [11–15]. At the same time, many deep learning-based object detection models have emerged for agricultural harvesting robots. For example, Xie Jiaxing et al. proposed a model called YOLOv5-litchi that detects lychees in natural environments by using an attention mechanism and increasing the small object detection layer, achieving an mAP@0.5 of 87.1% [16]. Zhipeng Cao et al. presented a real-time mango detection model based on YOLOv4 which improved the model detection speed by adjusting the network's width and depth and deleting some convolutional layers; this achieved an mAP@0.5 of 95.12% [17]. Jinhai Wang et al. utilized the Swin Transformer and DETR models to achieve grape bunch detection [18].

Deep learning-based object detection algorithms are mainly divided into two stages: two-stage detection and one-stage detection. Two-stage detection is based on candidate box detection algorithms, and some of the representative algorithms include the R-CNN seriesp [19], SPPNet [20], Fast R-CNN [21], and Faster R-CNN [22]. On the other hand, one-stage detection is simpler compared to two-stage detection as it is based on regression detection algorithms, directly generating the location coordinates and class probability of the target. One-stage detection has a lower training difficulty and a faster detection speed, and the YOLO series [23–26] is a typical representative of one-stage detection algorithms. Currently, the YOLOv5 algorithm is mostly applied for the detection of fruits such as apples [27], cherries [28], and tomatoes [29]; there are fewer studies on the automatic detection of Zanthoxylum.

In summary, the feature extraction-based methods used in previous studies to identify Zanthoxylum place high demand on the dataset, resulting in low detection accuracy when facing complex backgrounds and lighting conditions in natural environments; significant occlusion among Zanthoxylum branches, leaves, and fruits; and irregular shapes of each fruit on the Zanthoxylum spike. To achieve the detection of mature Zanthoxylum and assist the Zanthoxylum-harvesting robot in building a visual detection and positioning system, in this paper, a red-ripe Zanthoxylum image dataset is constructed. To enhance the directed feature extraction, the YOLOv5 model is used, and the CA (coordinate attention) mechanism is introduced to weaken the feature extraction of complex backgrounds. To specifically solve the problems of irregular Zanthoxylum spike shapes, complex field backgrounds, and dense Zanthoxylum fruits, the deformable convolution is introduced to improve the accuracy of mature Zanthoxylum recognition under natural conditions. In the second year, we deployed the model onto the platform constructed for the Zanthoxylum-harvesting robot for field tests in order to evaluate its effectiveness and generalizability.

## 2. Materials and Methods

### 2.1. Mature Zanthoxylum Image Collection

The images of mature Zanthoxylum were captured in Zijing Village, Shima Town, Boshan District, Zibo City, Shandong Province, from the Zanthoxylum plantations of local farmers. The images were collected on 25 August 2021 using a DJI motion camera, a Sony 5T camera, and a mobile phone in the natural environment.

The resolution of each collected image was 1280 × 1024 (pixels) with a 4:3 aspect ratio, and the original images were saved in JPG format. The images should include cases with single mother trees, multiple mother trees, Zanthoxylum branches, Zanthoxylum leaves,

and Zanthoxylum alone. In the natural environment, Zanthoxylum often has shading and backlighting, so when taking pictures with a camera, cases with shading and backlighting should be included as often as possible.

The growth and distribution of ripe Zanthoxylum on the Zanthoxylum tree in its natural environment are illustrated in Figure 1,. It can be seen from the figure that under natural conditions, the growth direction and the number of fruits in each cluster of Zanthoxylum are often not regular. The cluster of Zanthoxylum is a discrete target with an irregular shape. Under natural lighting conditions, shading and back-light are inevitable, making the color characteristics of Zanthoxylum unreliable. The overlap of branches and leaves in Zanthoxylum also results in an incomplete shape of the collected Zanthoxylum.

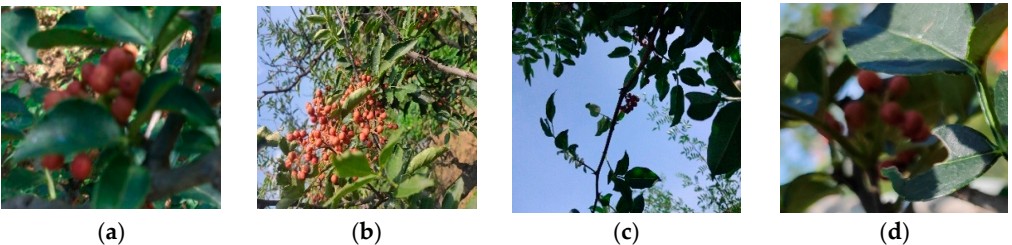

(**a**)          (**b**)          (**c**)          (**d**)

**Figure 1.** Images of Zanthoxylum peppercorn fruit under natural conditions. (**a**) Occlusion situation; (**b**) multi-mother plant overlap; (**c**) backlit conditions; (**d**) shading conditions.

### 2.2. Construction of the Dataset

To construct a deep learning model for the effective detection of Zanthoxylum under natural environmental conditions, this study only screened and removed images that were excessively blurred due to the shooting equipment not being completely focused. A total of 2827 images were collected at the trial site, and after screening and removal, 2368 images remained. The images were randomly divided into a training set, a validation set, and a test set in a 7:1:2 ratio. The dataset was annotated using the LabelImg annotation tool. The mature Zanthoxylum plants were selected by using a mouse to create a rectangular bounding box around the outer edge of the target contour, forming a quadrilateral bounding box. This study annotated the irregularly shaped Zanthoxylum clusters, with no requirement for the size of the quadrilateral. The area of the quadrilateral bounding box was kept as close as possible to the area of the Zanthoxylum it contained. The blue rectangle shows the ripe prickly ash fruit. An example of the annotated sample is shown in Figure 2.

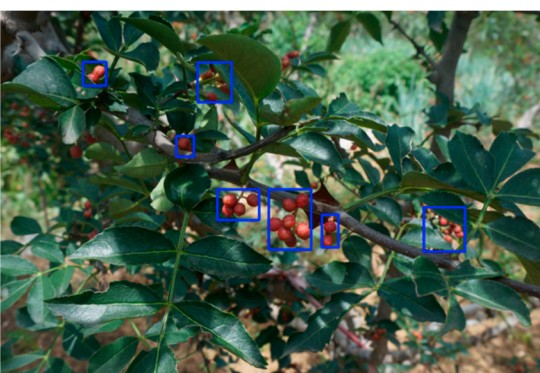

**Figure 2.** Sample image annotation.

To enhance the robustness of the object detection model while taking into account the tilt angle, illumination intensity, and different resolutions that may exist in the image acquisition process of field equipment, the original images used for modeling were augmented using methods such as geometric transformation, color transformation, and mixed

transformation. Ultimately, 12,000 images were obtained. The data augmentation examples are shown in Figure 3.

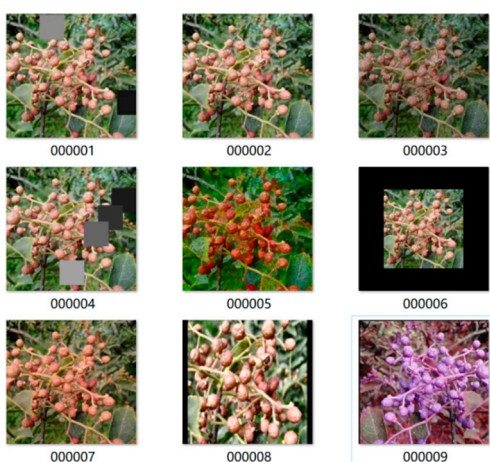

**Figure 3.** Data enhancement sample.

*2.3. Network Model Construction*

2.3.1. YOLOv5

YOLO is a representative of one-stage object detection algorithms which views object detection as a regression problem and performs feature extraction, object classification, and boundary box regression in a deep neural network, realizing end-to-end inference. It has a fast detection speed and can detect and classify objects in an image simultaneously.

The Zanthoxylum detection method, based on deep learning YOLO, can locate Zanthoxylum using real-time video and return its coordinates, category, and confidence. In the YOLO neural network, the input data are represented by an image, which is divided into S × S grids. When the center of the Zanthoxylum falls into a grid, the grid will detect it. Each grid detects B targets, and each target will receive 5 prediction parameters: x, y, w, h, and confidence, where (x, y) represent the target's coordinates and (w, h) represent the width and height of the boundary box.

YOLOv5 has four main parts in its network structure: the input end, backbone network, neck network, and output end. The input end represents the input image and includes some image preprocessing, including resizing the input image to the input size of the network and normalizing it. The backbone network of YOLOv5 uses Focus [30] as the benchmark network, which mainly uses slicing operations to crop the input image. In the neck portion, YOLOv5 adopts the fast spatial pyramid pooling [31] (SPPF) module for multi-scale feature fusion, as well as the feature pyramid network (FPN) [32] and the path aggregation network PAN [33] modules for network feature fusion and strengthening. The output end is used to output the object detection results.

2.3.2. A YOLOv5 Model Incorporating Attention Mechanisms and Deformable Convolutions

The introduction of attention mechanisms in deep learning networks can enhance the interested target region. Deformable convolution kernels can be adjusted based on the actual size and shape of the detected target, thus more effectively extracting the features of the detected object. As the shapes and sizes of mature chili pepper fruit spikes are irregular, to improve the detection accuracy of mature chili pepper fruit, this paper incorporates the attention mechanism module and the deformable convolution module into the YOLOv5 network.

Coordinate attention (CA) [34] is a kind of attention mechanism proposed by Qibin Hou et al. in 2021. The mechanism embeds position information into channel attention. The module decomposes channel attention into a 1D feature-decoding process in which features

are aggregated along different directions. During this process, the long-range features are extracted along one spatial direction, and precise position information is retained along the other spatial direction. The resulting feature maps are then encoded and aggregated to produce position- and direction-sensitive feature maps, thus enhancing the interest target area [35]. The specific structure of the CA attention mechanism module is shown in Figure 4.

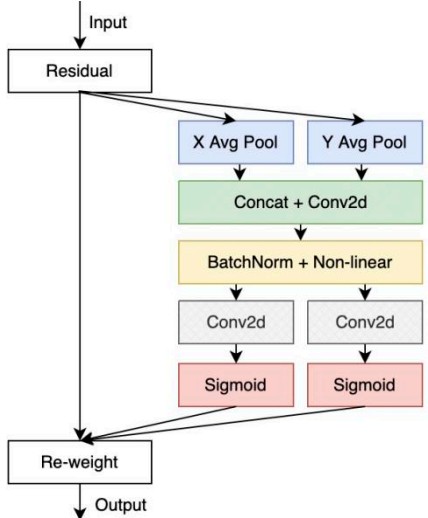

**Figure 4.** Structure diagram of the coordinate attention mechanism.

The detection of mature Zanthoxylum is highly correlated with the region of the fruit in the image. The model's sensitivity to the position of mature Zanthoxylum helps to improve detection accuracy.

In this paper, the model was required to be deployed on mobile devices. The early stage of the network is focused on shallow features, and adding the CA module during this stage would decrease the training and detection speed due to the high number of features considered in this stage. Therefore, the CA module was added before the SPPF module.

Deformable convolutional networks (DCNs) [36] are novel convolutional methods introduced by Dai et al. in 2017. The deformable convolution adds a direction offset to each element of the convolutional kernel, allowing the kernel to adjust its shape according to the actual object being detected and to better extract the input features. This type of convolution captures local features more effectively, especially when the object shape changes, making the advantages of deformable convolution more apparent. Figure 5 shows a comparison between a conventional 3 × 3 convolutional kernel and deformable convolution.

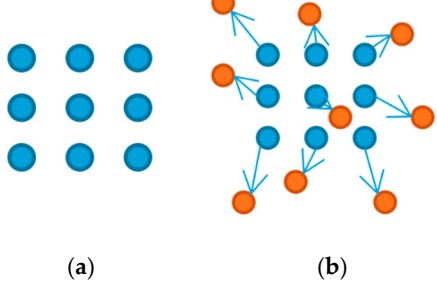

(**a**)　　　　　　　　　　　　　　　　(**b**)

**Figure 5.** Comparison of conventional 3 × 3 convolution and deformable convolution. (**a**) Normal convolution; (**b**) deformable convolution.

DCN can improve the model's ability to extract features from objects with deformation. The offset is learned by parallel convolutional layers, and the kernel can be shifted at the

sampling points on the input feature map. This causes the model to focus on the target area for detection and on making the kernel shape more suitable for the target shape, rather than being limited to a square sampling area. However, the first generation of deformable convolution may extend beyond the target area of interest and cause performance degradation, so Deformable ConvNets v2 (DCNv2) [37,38] introduced the addition of weight to each sampling point while learning the offset. This not only enhanced the acceptance of the input feature position, but also regulated the amplitude of the input feature, thus increasing the model's ability to model and learn. The following is the operation flow of the DCNv2 module.

Initially, if a $3 \times 3$ convolutional kernel is adopted, the definition of the kernel is R, and the size of the kernel is two-dimensional, as shown in Formula (1).

$$R = \{(-1, 1), (0, 1), \ldots, (0, 1), (1, 1)\} \tag{1}$$

DCNv2 first extracts the feature map using conventional convolution kernels, and then takes the obtained feature map as input, applying another convolution layer to the feature map to obtain the deformable convolution offset. The calculation formula of the normal convolution operation's output feature map is shown in Formula (2).

$$y(p_0) = \sum_{p_n \in R} w(p_n) \cdot x(p_0 + p_n) \tag{2}$$

In the formula, $p_0$ is the center point of the conventional convolution kernel; $p_n$ is the sampling point of the conventional convolution kernel; x is the input feature map; and y is the output feature map.

The formula for calculating the output feature map using a deformable convolution kernel is shown in Formula (3).

$$y(p_0) = \sum_{p_n \in R} w(p_n) \cdot x(p_0 + p_n + \Delta p_n) \Delta m_n \tag{3}$$

In this formula, $\Delta p_n$ represents the adjusted offset, $\Delta m_n$ represents the weight coefficient, and the remaining variables are the same as those in the traditional convolution operation. The deformable convolution introduces the position offset of the sampling points on the basis of the traditional convolution, which enables the output feature map to better represent the features of irregular targets. The offset $\Delta p_n$ shifts the points in region R based on the distribution of target features, and since the offset is generated by convolving the input feature map with another convolution layer, it is usually represented by a decimal. Therefore, by performing bilinear interpolation on the offset, the formula of the deformable convolution is transformed into Formula (4).

$$X(p) = \sum_q G(q, p) \cdot x(q) \tag{4}$$

In Equation (4), q represents the position of the sample point after being offset, p represents the integer grid point, and G (q, p) represents the integer form of the sample point position obtained from the bilinear interpolation operation. The structure diagram of deformable convolution is shown in Figure 6.

The model proposed in this paper needs to be deployed on mobile devices, so the YOLOv5s model with the smallest number of parameters was selected for improvement. Figure 7 shows the structure of the improved YOLOv5 network model. The CA attention mechanism module was inserted before the SPPF module of the backbone network in this model, and the deformable convolution module was introduced into the neck of the model.

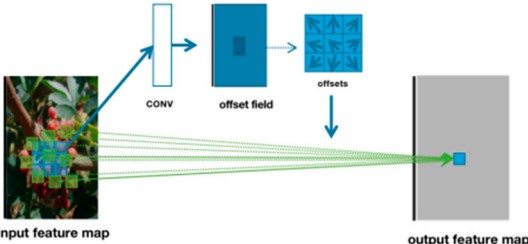

**Figure 6.** Deformable convolutional structure.

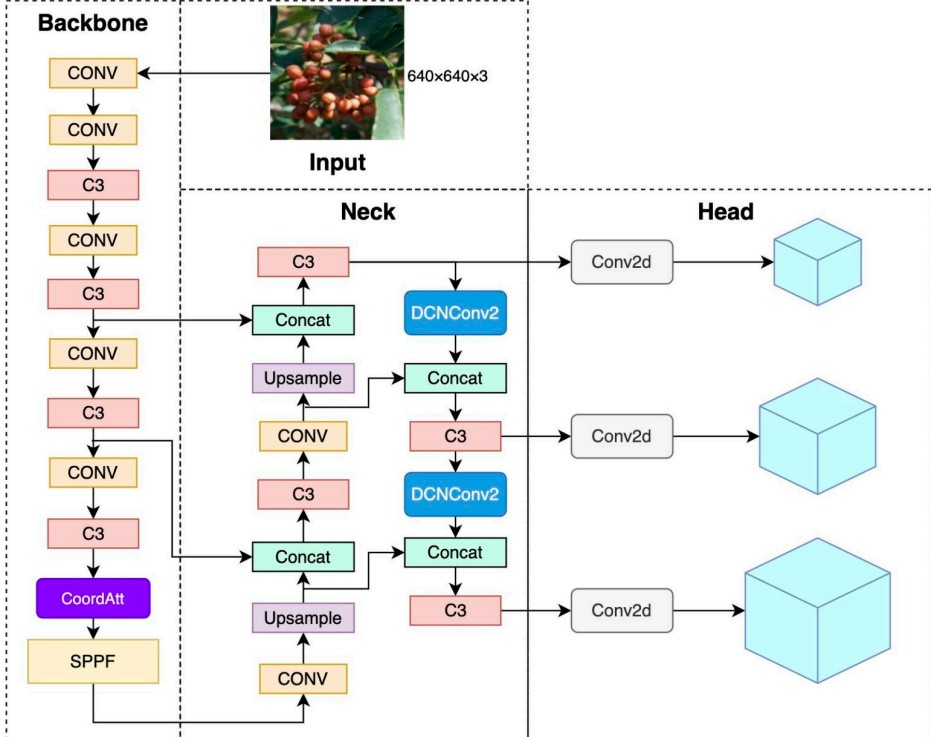

**Figure 7.** Improved YOLOv5 network map. Conv represents convolution, C3 represents a module consisting of three Convs and multiple bottleneck layers, SPPF refers to a spatial pyramid pooling-fast structure, Concat represents a feature fusion method of channel connection, Upsample represents up-sampling, and DCNConv2 refers to the deformable convolution module.

## 3. Experimental Design

### 3.1. Model Training

#### 3.1.1. Model Training Parameters

The platform for training and testing the model in this paper was a workstation computer with an Intel Core (TM) i9-9820X processor, operating at a frequency of 3.3 GHz, with 32 GB running memory and a GeForce GTX 2080ti GPU with 11 GB of memory. The operating environment was Ubuntu18.04 LTS. The training and testing of the model were based on the Pytorch framework, using the Python programming language and libraries such as CUDA, Cudnn, and OPENCV for setup.

The input image size for model training was 640 pixels by 640 pixels, and the model was trained for a total of 200 epochs. In order to evaluate the performance of the model, the weights parameters were saved after each epoch. The learning rate was warmed up using a warm-up method, and during this stage, the learning rate was updated through linear interpolation followed by the use of the cosine annealing algorithm.

The loss function is a value that represents the level of agreement between the model's prediction and the truth. Its magnitude determines the performance of the model. During

the model training process, factors that affect the training accuracy include the box loss (box_loss), object confidence loss (obj_loss), and classification loss (cls_loss). The loss function for the model in this paper is defined as shown in Equation (5).

$$\text{Loss} = 0.3 \times \text{box\_loss} + 0.4 \times \text{obj\_loss} + 0.3 \times \text{cls\_loss} \tag{5}$$

The change curve of the model's loss value during the training process is shown in Figure 8. From the figure, it can be seen that the loss value rapidly decreased in the first 15 rounds of training. After 100 rounds of training, the loss value was basically stable; the loss of the training set and the validation set have converged, and the gap between them is very small. The model did not show overfitting.

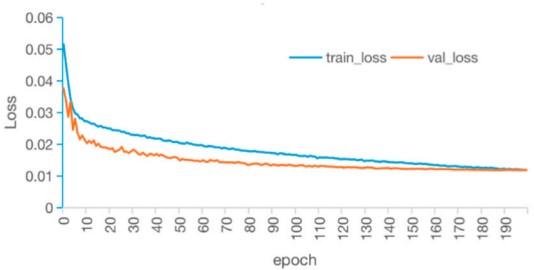

**Figure 8.** Change curve of loss value.

3.1.2. Model Evaluation Index

In this study, we primarily evaluated the performance of the output model using precision (P), recall (R), F1 score, mean average precision (mAP), and frame per second (FPS). The most intuitive metric for measuring the model's detection and classification ability is mAP, and the higher the accuracy of the model, the higher the mAP value. Therefore, we use the size of the mAP value as the primary evaluation criterion for the model. The intersection over union (IOU) is the ratio of the intersection and union of the generated candidate box and the original annotated box. mAP@0.5 indicates that the average precision mean is calculated when the IOU threshold is set to 0.5, and mAP@0.5:0.95 indicates the average value of mAP at different IOU thresholds (from 0.5 to 0.95 with a step of 0.05). The Zanthoxylum detection algorithm proposed in this paper is intended for use in the visual detection system of an intelligent Zanthoxylum-picking robot, which requires certain accuracy in terms of locating mature Zanthoxylum, so we evaluated the model using both mAP@0.5 and mAP@0.5:0.95.

Precision, denoted as P, is the ratio of the number of accurately predicted samples to the total number of samples, and its formula is as follows:

$$P = \frac{T_P}{T_P + F_P} \times 100\% \tag{6}$$

$T_P$ is the number of positive samples that were correctly predicted as positive, and $F_P$ is the number of negative samples that were wrongly predicted as positive. R is the proportion of all positive samples that were correctly predicted as positive, and is calculated as follows:

$$R = \frac{T_P}{T_P + F_N} \times 100\% \tag{7}$$

where $F_N$ is the number of positive samples that were wrongly predicted as negative.

F1 score is a metric that balances precision and recall, and is calculated as the harmonic mean of precision and recall. The formula is as follows:

$$F1 = 2 \times \frac{P \times R}{(P + R)} \tag{8}$$

The average precision (AP) is the area under the P-R curve, with recall R as the *X*-axis and precision P as the *Y*-axis. The mean average precision (mAP) is the average of the AP values for each category, obtained by summing up the AP values of each category and dividing by the total number of categories. The formula for this calculation is as follows:

$$AP = \int_0^1 P(r)dr \tag{9}$$

$$mAP = \frac{\sum_{i=1}^{N} AP_i}{N} \tag{10}$$

where P(r) represents the expression of the function for the P-R curve, N denotes the number of categories, and $AP_i$ represents the average precision value for category i.

### 3.2. Test Platform Construction

The Zanthoxylum detection algorithm proposed in this paper was intended to be applied to the visual system of a smart Zanthoxylum-picking robot. To test the practicality and problems of the algorithm, a Zanthoxylum-picking robot platform was set up as shown in Figure 9.

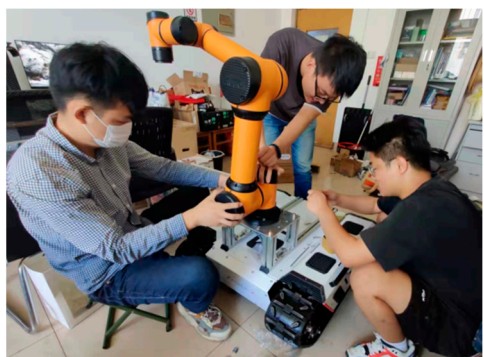

**Figure 9.** Construction of test platform.

The platform consisted of a tracked chassis and a six-axis robotic arm. The trained detection model was embedded within an industrial control computer and mounted on a D435i depth camera for image acquisition and the detection of ripe Zanthoxylum on trees. The software and hardware parameters of the industrial control computer are listed in Table 1.

**Table 1.** Hardware and software parameters of industrial computer.

| Name | Parameter |
|---|---|
| CPU | I7-1165G7 |
| Memory | 16GB |
| GPU | RTX2060-6GB |
| System | Ubuntu18.04 |
| Python version | 3.8.13 |
| Pytorch version | 1.12.0 |

To verify the effectiveness and generalizability of the trained model, it was deployed on the experimental platform, and a field test for Zanthoxylum detection was conducted in Zijing Village, Shima Town, Boshan District, Zibo City, Shandong Province, on 29 September 2022. The performance of the Zanthoxylum detection model was tested in real-world scenarios, as shown in Figure 10.

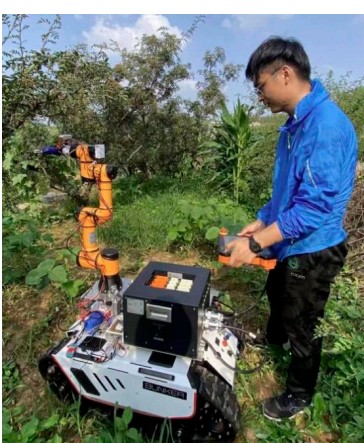

**Figure 10.** Peppercorn-picking robot test platform.

## 4. Results

*4.1. Comparative Analysis of Algorithm Optimization Experiment and Results*

Table 2 lists the different models used in this study, along with their corresponding descriptions. The models were trained and tested using the same dataset.

**Table 2.** Model name and comparison description.

| Number | Models | Explain |
|:---:|:---:|:---:|
| 1 | YOLOv5 | YOLOv5s |
| 2 | All-DCNv2-YOLOv5 | The convolutional layers in the backbone network are all replaced with deformable convolutions |
| 3 | CA-YOLOv5 | The CA module is added to the backbone network |
| 4 | DCNv2-YOLOv5 | The neck network introduced DCNv2 |
| 5 | CA-DCNv2-YOLOv5 | The backbone network adds the CA, and the Neck network introduces the DCNv2 |
| 6 | Faster R-CNN | A typical two-stage detection algorithm |
| 7 | SSD [39] | A typical one-stage detection algorithm |
| 8 | CenterNet [40] | A typical one-stage detection algorithm |

### 4.1.1. Ablation Study

In order to demonstrate the effectiveness of the proposed CA-DCNv2-YOLOv5 model, an ablation study was designed to verify the impact of different usage methods of the CA and DCNv2 on the model's performance in terms of detecting ripe Zanthoxylum.

A comparison of the changes in accuracy, recall, mAP@0.5, and mAP@0.5:0.95 during the 200-round training processes of different improved YOLOv5 algorithms is shown in Figure 11.

As shown in Figure 11, it can be seen that simply replacing the conventional convolution in the backbone network with deformable convolution modules had a limited ability to improve the model's performance, and caused relatively severe oscillation in the first half of the model training. Additionally, as each deformable convolution module required separate calculation of the offset, the computation of the model was increased, leading to an increase in both the training and detection times of the model. The results of the ablation comparison experiment are shown in Table 3.

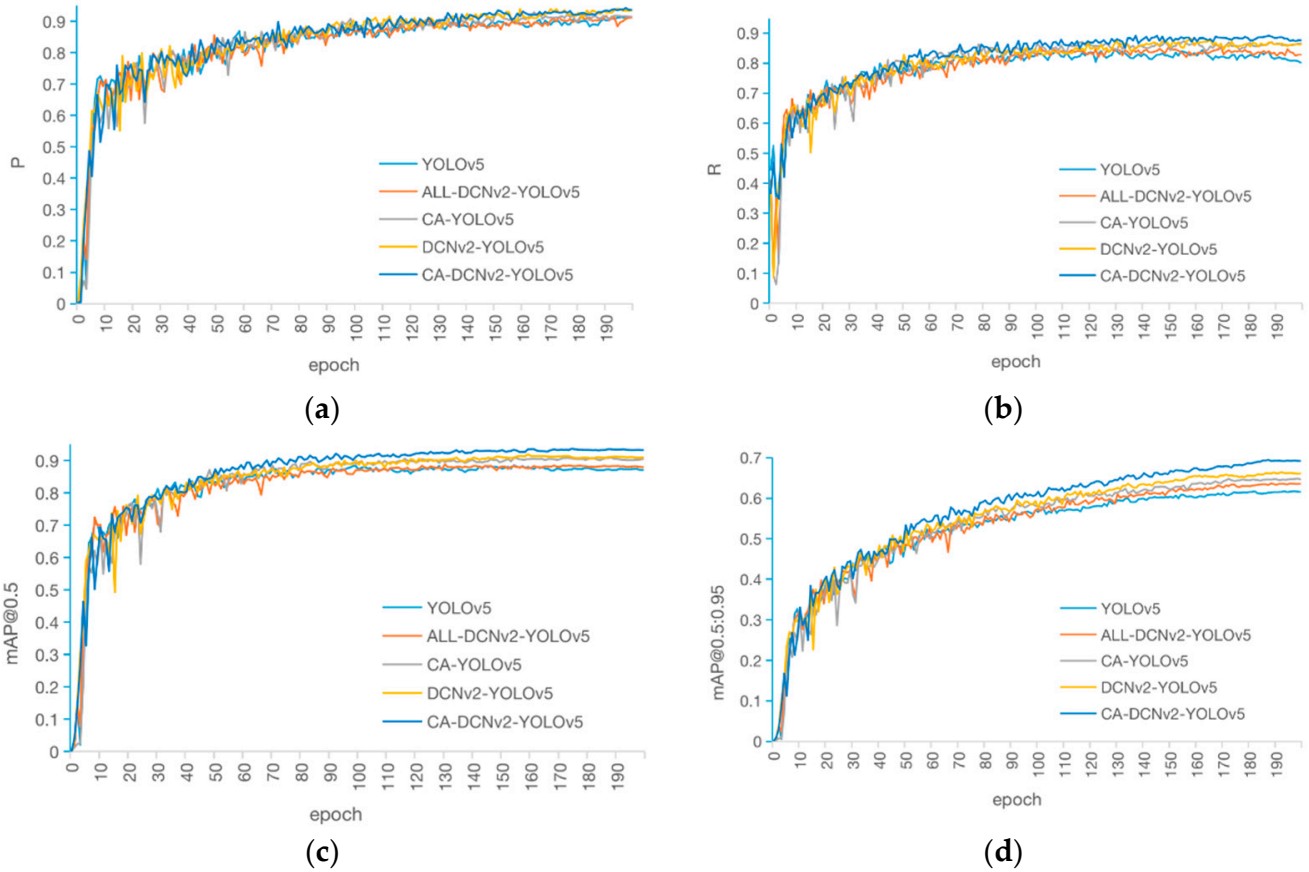

**Figure 11.** Evaluation index changes of different YOLO algorithms in the training process. (**a**) The precision change curve; (**b**) the recall change curve; (**c**) the change curve of mAP@0.5; (**d**) the change curve of mAP@0.5:0.95.

**Table 3.** Ablation comparison experiment results.

| Number | F1 Score | mAP@0.5/% | mAP@0.5:0.95/% | Speed/(Frame/s) | Model Size/(M) |
|--------|----------|-----------|----------------|-----------------|----------------|
| 1 | 0.86 | 88.9 | 62.6 | 97.1 | 14.5 |
| 2 | 0.87 | 88.3 | 63.6 | 82.7 | 14.7 |
| 3 | 0.89 | 90.4 | 64.8 | 97.1 | 14.8 |
| 4 | 0.90 | 91.1 | 66.4 | 95.2 | 14.6 |
| 5 | 0.91 | 93.5 | 69.5 | 95.2 | 14.7 |

Table 3 shows the results of the ablation study. The CA-YOLOv5 model, which incorporated the CA attention mechanism module, improved the mAP@0.5 and mAP@0.5:0.95 by 1.5% and 2.2%, respectively, compared to the original model. The DCNv2-YOLOv5 model, which introduced the deformable convolution modules, improved the mAP@0.5 by 2.2%, and the mAP@0.5:0.95 by 3.8%. However, simply replacing the conventional convolution modules in the backbone network with deformable convolution modules resulted in limited improvement to the model's accuracy and a substantial decrease in detection speed. The improved YOLO V5 model, which combined both improvements, further enhanced the detection accuracy of Zanthoxylum, with a 2.9% improvement in mAP@0.5 and a 2.9% improvement in mAP@0.5:0.95 compared to the CA-YOLOv5 model and the DCNv2-YOLOv5 model, respectively. Compared to the original YOLOv5 object detection model, the detection speed remained largely unchanged, with 4.6% and 6.9% improvements in mAP@0.5 and mAP@0.5:0.95, respectively.

From the above experiments, it can be seen that the introduction of the CA attention mechanism module and the proper replacement of the deformable convolution module

can effectively improve the target detection accuracy for mature Zanthoxylum. However, both combined in the CA-DCNv2-YOLOv5 model resulted in the best mAP@0.5 and mAP@0.5:0.95.

### 4.1.2. Comparison of Different Models

In addition to the ablation study, typical two-stage object detection algorithms, e.g., Faster R-CNN, and typical one-stage object detection algorithms, e.g., SSD and CenterNet, were trained using the dataset in this paper and tested with the same test set. The results are shown in Table 4.

**Table 4.** Performance comparison of different models.

| Model | F1 Score | mAP@0.5/% | mAP@0.5:0.95/% | Speed/(Frame/s) | Model Size/(M) |
|---|---|---|---|---|---|
| CA-DCNv2-YOLOv5 | 0.91 | 93.5 | 69.5 | 95.2 | 14.7 |
| Faster R-CNN | 0.85 | 85.9 | 55.9 | 16.0 | 113.4 |
| SSD | 0.76 | 79.6 | 40.6 | 98.9 | 95.5 |
| CenterNet | 0.69 | 77.8 | 38.5 | 81.1 | 131 |

The F1 score of the CA-DCNv2-YOLOv5 model was 0.91, with a mAP@0.5 of 93.5% and a mAP@0.5:0.95 of 69.5%, outperforming other network models. In terms of model size, that of the CA-DCNv2-YOLOv5 model was 14.7 M. On the other hand, that of the Faster R-CNN model was 113.4 M, which was close to 8 times the size of the CA-DCNv2-YOLOv5 model, and its detection speed was only one-sixth that of the CA-DCNv2-YOLOv5 model.

Comparing the four object detection algorithms, it can be seen that the CA-DCNv2-YOLOv5 model had the smallest model size, the highest F1 score, the highest mAP and the fastest detection speed. It achieved the best performance in detecting mature Zanthoxylum in natural environments.

### 4.1.3. Comparison of Model Detection Effect

A comparison of the different models' partial detection results can be seen in Figure 12. It can be seen that, except for Network 5, there were different degrees of false negatives, false positives, and repeated box selections in other models. However, the proposed Model 5 did not have these problems, and still showed good detection results for Zanthoxylum that was severely occluded by leaves. In addition, the size and position of the detection box were more accurate.

### 4.1.4. Network Attention Visualization

To demonstrate CA-DCNv2-YOLOv5's feature extraction capabilities more intuitively, this paper visualizes a feature heat map [41]. The results of model feature visualization are shown in Figure 13, where red areas indicate the regions on which the network is highly focused, with deeper colors indicating stronger levels of attention.

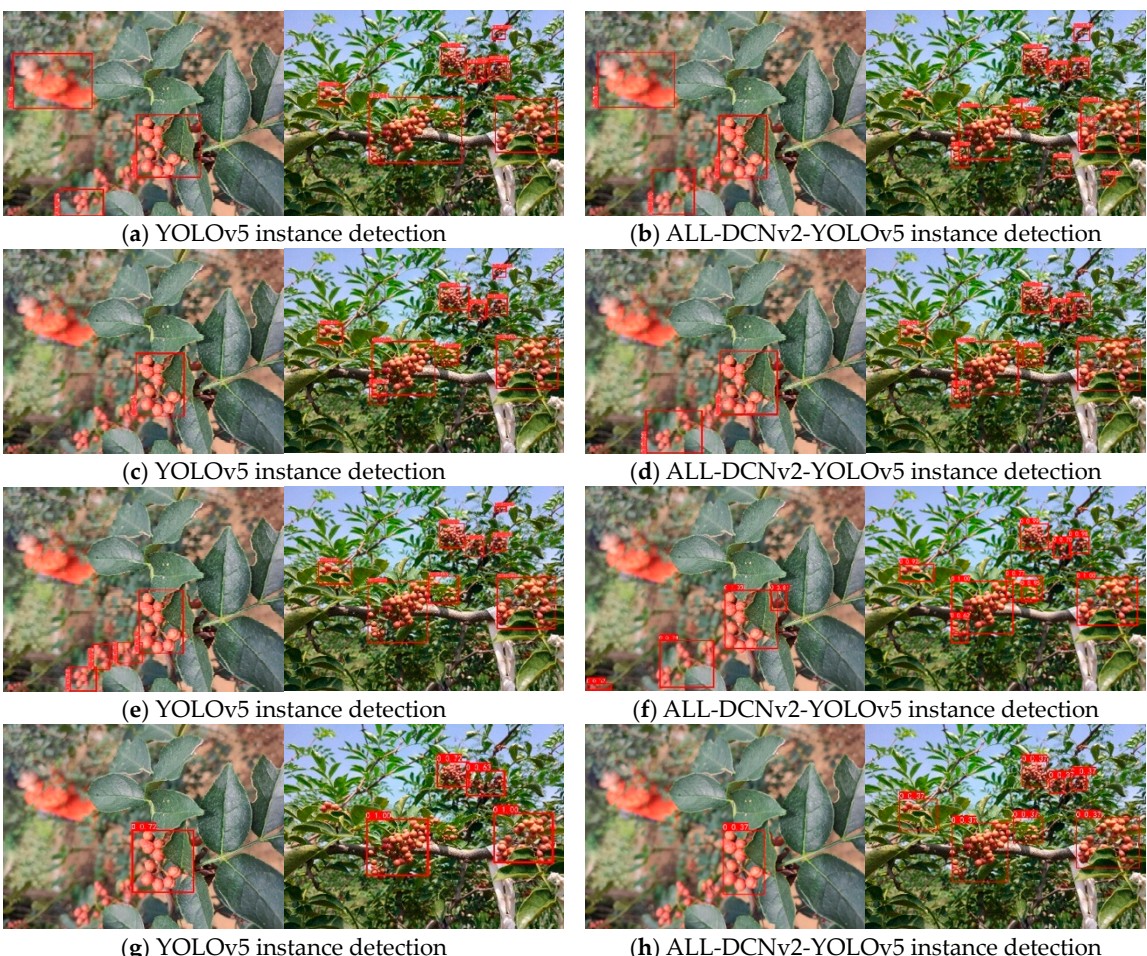

(**a**) YOLOv5 instance detection

(**b**) ALL-DCNv2-YOLOv5 instance detection

(**c**) YOLOv5 instance detection

(**d**) ALL-DCNv2-YOLOv5 instance detection

(**e**) YOLOv5 instance detection

(**f**) ALL-DCNv2-YOLOv5 instance detection

(**g**) YOLOv5 instance detection

(**h**) ALL-DCNv2-YOLOv5 instance detection

**Figure 12.** Comparison of partial test results of different models. (**a**) YOLOv5 instance detection; (**b**) ALL-DCNv2-YOLOv5 instance detection; (**c**) CA-YOLOv5 instance detection; (**d**) DCNv2-YOLOv5 instance detection; (**e**) CA-DCNv2-YOLOv5 instance detection; (**f**) Fsater R-CNN instance detection; (**g**) m SSD instance detection; (**h**) Centernet instance detection.

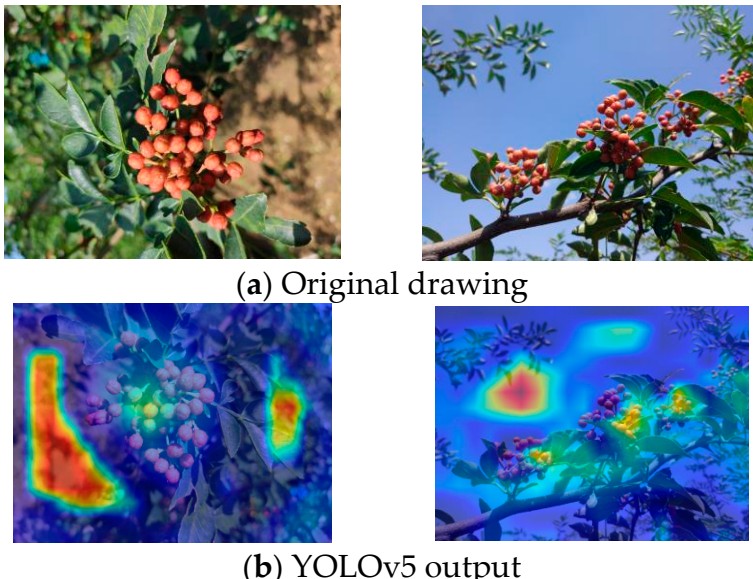

(**a**) Original drawing

(**b**) YOLOv5 output

**Figure 13.** *Cont.*

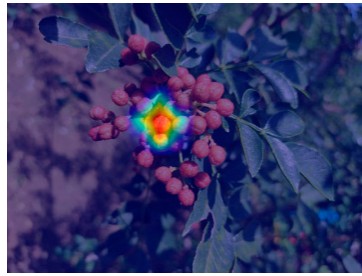 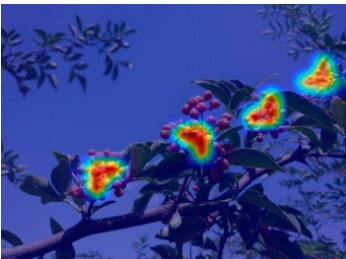

(**c**) CA-DCNv2-YOLOv5 Output

**Figure 13.** Class activation mapping.

It can be seen from the figure that compared to the YOLOv5 model, the CA-DCNv2-YOLOv5 model pays more attention to the local areas of the Zanthoxylum in the feature extraction process, and relatively less attention to irrelevant information. Thus, it showed higher accuracy in detecting mature Zanthoxylum.

4.1.5. Model Recognition Performance for Zanthoxylum under Occlusion Conditions

To further verify the detection performance of the improved CA-DCNv2-YOLOv5 model on occluded Zanthoxylum, a separate test set was constructed by manually selecting 100 images with occlusions from the test set, and both YOLOv5 model and CA-DCNv2-YOLOv5 model were used for prediction. The prediction results are shown in Table 5.

**Table 5.** Comparison of the detection effect of the occluded target fruit before and after the improvement.

| Models | mAP@0.5/% | mAP@0.5:0.95/% |
|:---:|:---:|:---:|
| YOLOv5 | 86.5 | 58.9 |
| CA-DCNv2-YOLOv5 | 91.9 | 63.6 |

As shown in Table 5, the CA-DCNv2-YOLOv5 model outperformed the YOLOv5 model in the test set with occlusions, with improvements of 5.4 and 4.7 percentage points in mAP@0.5 and mAP@0.5:0.95, respectively. These results demonstrate that the improved YOLOv5 target detection algorithm proposed in this paper helps to increase the detection accuracy of mature Zanthoxylum with occlusions.

*4.2. Field Experiment*

The feasibility and practicality of the proposed Zanthoxylum detection algorithm were demonstrated through the collection, recognition, and location of Zanthoxylum images. These images were obtained from different positions on trees in their natural environment by means of a mechanical arm in different poses and the improved YOLOv5 model. The results of the actual performance tests of the model are shown in Figure 14. The model was able to detect and recognize mature Zanthoxylum in the field and output the coordinate information, with an average detection time of 11.2 ms and a detection speed of 89.3 frames per second, thus satisfying the real-time detection requirements. The recognition and detection information can be used in real time to drive the Zanthoxylum-harvesting robot to perform the cutting, grasping, and collection tasks. The harvesting performance of the robot is shown in Figure 15.

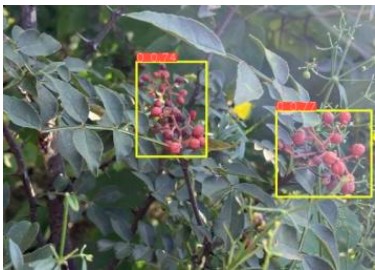
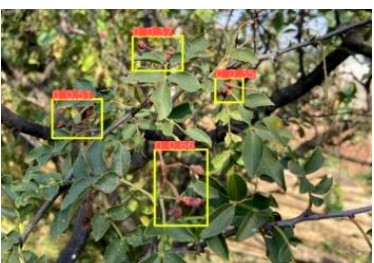

**Figure 14.** Detection effect of ripe prickly ash fruit in the field.

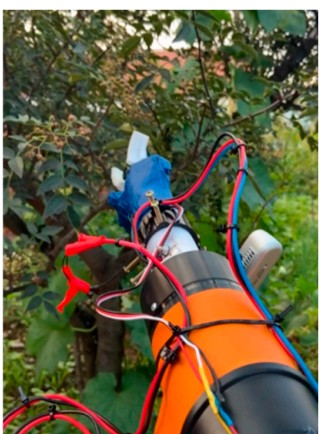
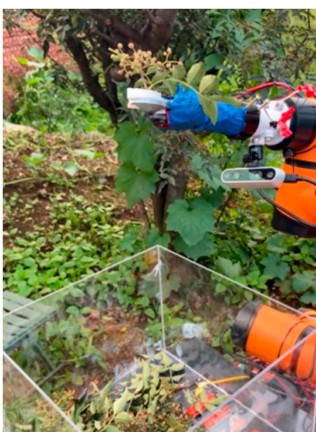

**Figure 15.** Prickly ash-picking robot work diagram.

## 5. Discussion

Zanthoxylum features a growth pattern in which the fruits are discrete, with an irregular spike shape. This makes the detection of the fruit challenging due to cross-occlusion between fruits. To address this challenge, this paper introduces a deformable convolutional module to better adapt to the shape of the Zanthoxylum and extract more features. At the start of the network, the feature map has a large number of features. Adding the deformable convolutional module at this point will cause the model to learn a significant number of irrelevant features and, thus, considerably reduce both the training speed and the detection speed. The proposed model is deployed on a mobile device, and has a certain requirement for detection speed. The experimental results showed that introducing the deformable convolutional module into the neck of the model instead into the backbone network significantly improves the detection speed. Furthermore, the introduction of a CA attention mechanism into the backbone network increased the model's sensitivity to positional information. Combining this with the deformable convolution improves the accuracy of the model in detecting mature Zanthoxylum.

Zanthoxylum undergoes a red maturation process that takes place over a period of time, approximately two months, during August and September. During this time, all Zanthoxylum is in the red maturation stage. Images of red mature Zanthoxylum collected during different months are shown in Figure 16. Figure 16a shows red mature Zanthoxylum from August, with more plump and fresh red fruit. Figure 16b shows mature red Zanthoxylum from September, the fruits of which are relatively dry and have a deep red color. In subsequent algorithmic improvements, the changes in color and fruit shape during the red maturation process of Zanthoxylum should be fully considered in order to further improve the robustness and detection accuracy of the algorithm.

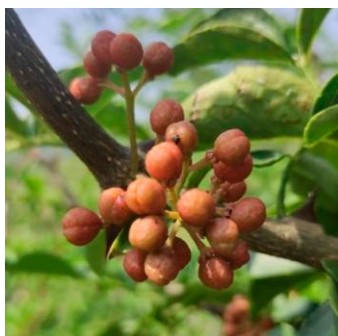 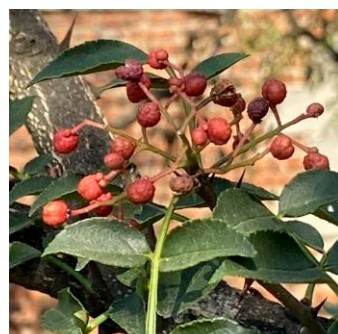

(**a**) Red ripe pepper fruit in August    (**b**) September red ripe prickly ash fruit

**Figure 16.** Comparison of red ripe prickly ash at different growth stages.

## 6. Conclusions

The visual detection system is a key module for the Zanthoxylum-harvesting robot. In order to achieve accurate detection of mature Zanthoxylum this paper presents an improved YOLO algorithm to detect Zanthoxylum in natural environments. The main conclusions are as follows:

1. An improved YOLOv5 model was proposed for Zanthoxylum cluster detection in its natural environment by adding the CA attention mechanism module into the backbone network and introducing the deformable convolutional module into the neck. The testing results showed that the improved model had an average accuracy of 93.5% in mAP@0.5 and 69.5% in mAP@0.5:0.95, which improved by 4.6% and 6.9%, respectively, compared to the original YOLOv5 model, while maintaining the basic detection speed. In addition, the CA-DCNv2-YOLOv5 model proposed in this paper demonstrated a significant performance advantage compared to Faster R-CNN, SSD, and CenterNet.
2. The improved YOLOv5 network model was tested on an image dataset that included occlusions of Zanthoxylum. The average precision scores, mAP@0.5 and mAP@0.5:0.95, improved by 5.4% and 4.7%, respectively, compared to the original YOLOv5 network.
3. The improved YOLOv5 network model had a detection speed of approximately 89.3 frames per second on mobile devices, meeting the real-time detection requirements of the Zanthoxylum-harvesting robot.

**Author Contributions:** Conceptualization, J.G. and J.Z.; methodology, J.G.; software, J.G.; validation, J.G., X.X. and J.M.; formal analysis, J.G.; investigation, J.G., J.Z. and B.T.; resources, Y.L.; data curation, J.G. and B.T.; writing—original draft preparation, J.G.; writing—review and editing, J.Z.; visualization, J.Z. and J.G.; supervision, J.Z.; project administration, J.Z. and Y.L.; funding acquisition, J.Z. and Y.L. All authors have read and agreed to the published version of the manuscript.

**Funding:** This study was funded by the "One Case, one Discussion" special fund for the introduction of top talents in Shandong Province (Lu Zhengban Zi [2018] No.27), supported by the Natural Science Foundation of Shandong Province (ZR2021MD091).

**Institutional Review Board Statement:** Not applicable.

**Data Availability Statement:** Not applicable.

**Conflicts of Interest:** The authors declare no conflict of interest.

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
