# Peer review of "Design and Experiment of a Visual Detection System for Zanthoxylum-Harvesting Robot Based on Improved YOLOv5 Model"

_agriculture, doi:10.3390/agriculture13040821_

Round 1

Reviewer 1 Report

The work presented in this paper describes a method for the detection of  a specific plant, while they also present the construction of a robot platform utilizing this method to pick the plant in reality. The paper is well-structured, understandable and describes the research adequately. While the theoretical novelty is average, it is nice to see a method that is actually applied in a real-world application. The application itself is rather narrow in scope, although the presented method can presumable be applied to other plants quite easily.

I have the following specific comments for corrections:

1. Figures 1. and 12. are outside the paper

2. Fig 13. the caption seems odd. Are the authors sure it is not a placeholder?

3. I find it somewhat strange that the entire construction from dataset to architecture to training to implementing the robot model is squeezed into a single section (section 2). It would be beneficial for the paper's structure to separate these into multiple sections.

4. Since the authors use an RGB-D camera, would it be possible/beneficial to add depth information as an additional input channel to yolo? If so, why did the authors decide not to do so?

5. Is there a video available of the robot in operation? If yes, it would be nice to include a link to it in the paper.

Author Response

Original Manuscript ID:  agriculture-2249417

Original Article Title: “Design and Experiment of a Visual Detection System for Zan-thoxylum Harvesting Robot Based on Improved YOLOv5 Model”

To: agriculture

Re: Response to reviewers

Dear Editor,

Thank you for allowing a revision of our manuscript, with an opportunity to address the reviewer’s comments. We have studied reviewer’s comments carefully and have made revisions which marked with revised format in the manuscript.

We are uploading (a) our point-by-point response to the comments (below) (response to reviewers), (b) an updated manuscript with red highlighting in+dicating changes (PDF document).

Point 1: Figures 1. and 12. are outside the paper.

Response 1:Thank you for your suggestions. We have adjusted the size of the images to ensure that they do not exceed the page limit.

Point 2: Fig 13. the caption seems odd. Are the authors sure it is not a placeholder?

Response 2: Thank you for your suggestion. We have revised the image names in the manuscript, and the correct name for the figure is "Class Activation Mapping."

Point 3: I find it somewhat strange that the entire construction from dataset to architecture to training to implementing the robot model is squeezed into a single section (section 2). It would be beneficial for the paper's structure to separate these into multiple sections.

Response 3: Thank you for your suggestion. We have split the original Section 2 into Section 2 “Materials and Methods” and Section 3 “Experimental Design”.

Point 4: Since the authors use an RGB-D camera, would it be possible/beneficial to add depth information as an additional input channel to yolo? If so, why did the authors decide not to do so?

Response 4: The data used in this study was collected in July and August 2021, and the deep camera was purchased after this time. Therefore, only RGB image data was used in the model training process. Currently, we are reading the corresponding depth information at the predicted positions after predicting the RGB images. Thank you very much for your suggestion, and we will consider using depth information as an additional channel input for YOLO in our future work.

Point 5:  Is there a video available of the robot in operation? If yes, it would be nice to include a link to it in the paper.

Response 5: We did not record videos during the experiment in this study. The working photos of the picking robot are shown in Figure 15. Thank you for your suggestion. We will pay more attention to recording videos of the robot's operation in future experiments.

We would like to express our great appreciation to you and reviewers for comments on our manuscript. We treasure the opportunity and wish that you and the reviewers would acknowledge the revised manuscript. Please inform us if you think the manuscript needs to modify again. We can revise the manuscript till you and reviewers are satisfied. We are looking forward to hearing from you soon.

Best regards,

Jinkai Guo 1,2, Xiao Xiao 1,2, Jianchi Miao 1,2, Bingquan Tian 1,2, Jing Zhao1,2 * and Yubin Lan3*

Reviewer 2 Report

The title "Design and Experiment of a Visual Detection System for Zanthoxylum Harvesting Robot Based on Improved YOLOv5 Model"

In this paper, the authors present an efficient zanthoxylum harvesting robot that uses the YOLOv5 model to automatically identify mature Zanthoxylum. They have improved the YOLOv5 model by incorporating a coordinate attention module and a deformable convolution module. The authors have conducted comprehensive experiments to evaluate the performance of the proposed approach, which has shown promising results and outperforms other models.

Strengths:

1. The topic is interesting and practical, and the proposed approach has the potential to greatly reduce the workload in large-scale zanthoxylum harvesting. It is encouraging to see that the authors have already developed a prototype.

2. The paper is well-structured and easy to follow, with a clear description of the methodology.

3. The authors have conducted thorough experiments to validate the effectiveness of their approach, which is a significant strength.

Weaknesses:

1. There are minor typos in Table 1, specifically in the "PyTorch version" column.

2. I believe your approach would be much more efficient compare to manually harvesting. However, I am still curisous is there any data to show that how efficiency your approach compare to manually harvesting? 

Author Response

Original Manuscript ID:  agriculture-2249417

Original Article Title: “Design and Experiment of a Visual Detection System for Zan-thoxylum Harvesting Robot Based on Improved YOLOv5 Model”

To: agriculture

Re: Response to reviewers

Dear Editor,

Thank you for allowing a revision of our manuscript, with an opportunity to address the reviewer’s comments. We have studied reviewer’s comments carefully and have made revisions which marked with revised format in the manuscript.

We are uploading (a) our point-by-point response to the comments (below) (response to reviewers), (b) an updated manuscript with red highlighting in+dicating changes (PDF document).

Point 1: The topic is interesting and practical, and the proposed approach has the potential to greatly reduce the workload in large-scale zanthoxylum harvesting. It is encouraging to see that the authors have already developed a prototype.

Response 1:Thank you for your valuable feedback, suggestions, and encouragement. We will continue to conduct research and make improvements in the future.

Point 2: The paper is well-structured and easy to follow, with a clear description of the methodology.

Response 2: Thank you for your valuable feedback, suggestions, and encouragement. We will continue to conduct research and make improvements in the future.

Point 3: The authors have conducted thorough experiments to validate the effectiveness of their approach, which is a significant strength.

Response 3: Thank you for your valuable feedback, suggestions, and encouragement. We will conduct more targeted research and experiments in the future based on your suggestions.

Point 4: There are minor typos in Table 1, specifically in the "PyTorch version" column.

Response 4: Thank you for your suggestion. We have made the necessary corrections to the typos or misspellings in the manuscript.

Point 5:  I believe your approach would be much more efficient compare to manually harvesting. However, I am still curisous is there any data to show that how efficiency your approach compare to manually harvesting?

Response 5: Using Zanthoxylum picking robots to harvest Zanthoxylum can help solve problems such as high injury rates among picking personnel and labor shortages in agriculture. This paper focuses on the research of Zanthoxylum target detection in the vision system of the Zanthoxylum picking robot, and the deployment of the algorithm on the robot for feasibility testing and generalization validation. Thank you for your suggestions. We will compare the robot picking with manual picking from multiple perspectives in future work.

We would like to express our great appreciation to you and reviewers for comments on our manuscript. We treasure the opportunity and wish that you and the reviewers would acknowledge the revised manuscript. Please inform us if you think the manuscript needs to modify again. We can revise the manuscript till you and reviewers are satisfied. We are looking forward to hearing from you soon.

Best regards,

Jinkai Guo 1,2, Xiao Xiao 1,2, Jianchi Miao 1,2, Bingquan Tian 1,2, Jing Zhao1,2 * and Yubin Lan3*
